# Heterozygous missense variant of the proteasome subunit β-type 9 causes neonatal-onset autoinflammation and immunodeficiency

Nobuo Kanazawa 1,2,17✉, Hiroaki Hemmi3,4,17, Noriko Kinjo5, Hidenori Ohnishi 6, Jun Hamazaki7, Hiroyuki Mishima 8, Akira Kinoshita8, Tsunehiro Mizushima9, Satoru Hamada5, Kazuya Hamada5, Norio Kawamoto 6, Saori Kadowaki6, Yoshitaka Honda10, Kazushi Izawa 10, Ryuta Nishikomori 11, Miyuki Tsumura12, Yusuke Yamashita13, Shinobu Tamura13, Takashi Orimo3,14, Toshiya Ozasa3, Takashi Kato 3, Izumi Sasaki3, Yuri Fukuda-Ohta3, Naoko Wakaki-Nishiyama3, Yutaka Inaba 1, Kayo Kunimoto1, Satoshi Okada 12, Takeshi Taketani15, Koichi Nakanishi 5, Shigeo Murata 7, Koh-ichiro Yoshiura 8,16 & Tsuneyasu Kaisho 3✉

Impaired proteasome activity due to genetic variants of certain subunits might lead to proteasome-associated autoinflammatory syndromes (PRAAS). Here we report a de novo heterozygous missense variant of the PSMB9 proteasome subunit gene in two unrelated Japanese infants resulting in amino acid substitution of the glycine (G) by aspartic acid (D) at position 156 of the encoded protein β1i. In addition to PRAAS-like manifestations, these individuals suffer from pulmonary hypertension and immunodeficiency, which are distinct from typical PRAAS symptoms. The missense variant results in impaired immunoproteasome maturation and activity, yet ubiquitin accumulation is hardly detectable in the patients. A mouse model of the heterozygous human genetic variant (Psmb9G156D/+) recapitulates the proteasome defects and the immunodeficiency phenotype of patients. Structurally, PSMB9 G156D interferes with the β-ring-βring interaction of the wild type protein that is necessary for 20S proteasome formation. We propose the term, proteasome-associated autoinflammatory syndrome with immunodeficiency (PRAAS-ID), to indicate a separate category of autoinflammatory diseases, similar to, but distinct from PRAAS, that describes the patients in this study.

[1] Department of Dermatology, Wakayama Medical University, Wakayama, Japan. [2] Department of Dermatology, Hyogo College of Medicine, Nishinomiya, Japan. [3] Department of Immunology, Institute of Advanced Medicine, Wakayama Medical University, Wakayama, Japan. [4] Laboratory of Immunology, Faculty of Veterinary Medicine, Okayama University of Science, Imabari, Japan. [5] Department of Child Health and Welfare (Pediatrics), Graduate School of Medicine, University of the Ryukyus, Nishibara, Japan. [6] Department of Pediatrics, Graduate School of Medicine, Gifu University, Gifu, Japan. [7] Laboratory of Protein Metabolism, Graduate School of Pharmaceutical Sciences, The University of Tokyo, Tokyo, Japan. [8] Department of Human Genetics, Atomic Bomb Disease Institute, Nagasaki University, Nagasaki, Japan. [9] Graduate School of Science, University of Hyogo, Himeji, Japan. [10] Department of Pediatrics, Kyoto University Graduate School of Medicine, Kyoto, Japan. [11] Department of Pediatrics and Child Health, Kurume University School of Medicine, Kurume, Japan. [12] Department of Pediatrics, Hiroshima University Graduate School of Biomedical and Health Sciences, Hiroshima, Japan. [13] Department of Hematology/Oncology, Wakayama Medical University, Wakayama, Japan. [14] Laboratory of Immune Regulation, Department of Microbiology and Immunology, Graduate School of Medicine, Osaka University, Suita, Japan. [15] Department of Pediatrics, Shimane University Faculty of Medicine, Izumo, Japan. [16] Division of Advanced Preventive Medical Sciences and Leading Medical Research Core Unit, Nagasaki Univeristy Graduate School of Biomedical Sciences, Nagasaki, Japan. [17] These authors contributed equally: Nobuo Kanazawa, Hiroaki Hemmi. ✉email: nkanazaw@hyo-med.ac.jp; tkaisho@wakayama-med.ac.jp

V ariants of *proteasome subunit β-type 8 (PSMB8)* were detected in autoinflammatory diseases characterized by systemic relapsing inflammations and progressive wasting, such as Nakajo-Nishimura syndrome and chronic atypical neutrophilic dermatosis with elevated temperature (CANDLE)[1–4]. Subsequently, identification of mostly biallelic variants of other proteasome subunit and chaperone genes, leading to loss-of-function of the proteasome, has defined a disease entity as proteasome-associated autoinflammatory syndrome (PRAAS)[5–8].

The proteasome is a protein complex involved in intracellular protein homeostasis by degrading unnecessary or useless proteins tagged with polyubiquitin[9,10]. All eukaryotic cells have the constitutive 26S proteasome which consists of a 20S core particle and two 19S regulatory particles. The 20S core complex is composed of α1-α7 and β1-β7 subunits, among which β1, β2, and β5 mediate the protease activity. In hematopoietic cells or fibroblasts stimulated with cytokines, inducible subunits, β1i (coded by *PSMB9*), β2i (coded by *PSMB10*), and β5i (coded by *PSMB8*), are substituted for β1, β2, and β5, respectively, to form the immunoproteasome. Especially in thymic cortical epithelial cells, β5t (coded by *PSMB11*), instead of β5i, as well as β1i and β2i are expressed to form the thymoproteasome. Immunoproteasome and thymoproteasome are involved not only in protein degradation but also in generation of antigen peptides presented with major histocompatibility complex (MHC) class I molecules and CD8 T cell repertoire, population, and responses.

In this work, we identify a de novo *PSMB9* heterozygous missense variant, G156D, in two unrelated Japanese patients with manifestations, including autoinflammation and immunodeficiency, which are similar to, but distinct from those of PRAAS patients. The proteasome defect and immunodeficient phenotypes are recapitulated in *Psmb9*[G156D/+] mice.

## Results

**Clinical phenotype**. Two patients from nonconsanguineous families were enrolled in this study (Supplementary Note 1). These patients showed similar features, such as neonatal-onset fever, increased inflammatory reactants, skin rash (Fig. 1a, b, and Supplementary Figs. 1a and 2a, b), myositis (Fig. 1c, and Supplementary Fig. 1b), liver dysfunction, pulmonary arterial hypertension (Fig. 1d and Supplementary Fig. 2e, f) and basal ganglia calcification (Fig. 1e and Supplementary Fig. 2c). Autoantibodies associated with dermatomyositis were not detected in either patient. The serum inflammatory cytokine levels, such as IL-6, IL-18, and IP-10, were increased in both patients. Concerning immunological functions, patient 1 showed decrease of serum IgG level and B cell numbers (Supplementary Table 1), and required IgG supplementation to control manifestations. Patient 1 also exhibited low levels of both T-cell receptor recombination excision circles (TREC) and immunoglobulin (Ig) κ-deleting recombination excision circles (KREC), indicating T as well as B cell defects. In patient 1, during immunosuppressive treatments, interferon (IFN) score was not elevated, and signal transducers and activators of transcription 1 (STAT1) phosphorylation were scarce (Supplementary Fig. 1e, f). Meanwhile, patient 2 showed decrease of monocytes, CD8 T, and γδ T cells and almost absence and decreased activity of natural killer (NK) cells (Supplementary Table 1). In patient 2, interferon (IFN) -α was not only increased in the serum but also detected in cerebrospinal fluid. Furthermore, despite the usage of prednisolone the IFN score was elevated in the whole blood, and STAT1 phosphorylation was enhanced in the skin lesion collected. This was also shown in the experiments with dermal fibroblasts (Fig. 1f–h). Patient 2 also showed manifestations compatible with hemophagocytic

lymphohistiocytosis[11]. Interestingly, polyoma (BK/JC) viruses were detected in both patient's peripheral blood samples (Table 1). In patient 1, the viruses were detected at 4 and 10-year-old, when the effects of immunosuppressants could not be excluded, while in patient 2 they were detected at 3 days of age, at the time without any therapy.

The detailed case reports were described in Supplementary Note 1 and summarized in Table 1. In summary, these two patients initially showed mainly severe autoinflammatory phenotypes and later manifested the immunodeficiency phenotype with periodic inflammatory exacerbation. Clinical features of both patients were partially overlapped with PRAAS. However, they apparently lacked lipoatrophy, one of the main characteristics of PRAAS, and showed pulmonary hypertension, which is rare in PRAAS. More importantly, they also manifested combined immunodeficiency (at least B and T cells in patient 1 and T and NK cells in patient 2), which is not seen in PRAAS (Table 1).

**Genetic and functional analysis**. In patient 1, no pathogenic variant was identified in MEFV, MVK, NLRP3, NOD2, TNFRSF1A, or PSMB8. Then, a panel of ubiquitin-proteasome-system, autophagy and interferonopathy-related genes (Supplementary Table 2) were sequenced and heterozygous PSMB9 (NM_002800) c.467 G > A (p.G156D) was identified (Fig. 2a). This is a de novo variant in the immunoproteasome-specific β1i subunit gene, and 156 G is well conserved among β1 and β1i across species (Fig. 2c). Whole exome sequencing of genomic DNAs from the patient and their parents revealed two homozygous, one compound heterozygous, and 29 de novo variants in patient 1 (Supplementary Table 3). Among these genes, only *PSMB9* was associated with inflammation. In patient 2, a panel of autoinflammatory disease-causing genes (Supplementary Table 2) were analyzed and de novo *PSMB9* c.467 G > A (p.G156D) heterozygous variant, i.e. the same variant found in patient 1, was identified (Fig. 2b). Whole exome sequencing of genomic DNAs from the patient and their parents identified two homozygous, one compound heterozygous and 12 de novo variants in patient 2 (Supplementary Table 4). Among these genes, only *PSMB9* was associated with inflammation and the *PSMB9* variant was the only variant found in common between two patients. Although clinical phenotypes of the patients are similar to that of NEMO deleted exon 5-autoinflammatory syndrome (NEMO-NDAS), the possibility that they belong to NEMO-NDAS was excluded, because re-evaluation of the *NEMO* genes showed no variants[12].

Structural modeling revealed that G156 in β1i is located on the interface between two β rings and G156D substitution affects their interaction (Fig. 2d), while keeping the active site conformation (Supplementary Fig. 3a). Consistent with this modeling, immunoblotting analysis of patient 1-derived immortalized B cells showed increase of immunoproteasome intermediate containing Ump1 and immature β1i and severely diminished incorporation of all induced subunits into the 20S complex (Fig. 3a and Supplementary Fig. 3b). Meanwhile, in the 26S proteasome of patient 1-derived immortalized B cells, β2i and β5i incorporation was comparable to that of his mother-derived control cells, although β1i incorporation was impaired and immature β1i was detected. In the patient 1-derived cells, the catalytic activity of the 20S, but not the 26S proteasome, was severely impaired, whereas both the 20S and 26S proteasome showed severe defects in the activities in PRAAS patient-derived cells (Fig. 3b). Ubiquitin accumulation, which is caused by the 26S proteasome defect and is common in PRAAS, was unremarkable in the patients' skin lesion (Fig. 3c and Supplementary Fig. 4a). Patient 2-derived fibroblasts exhibited a peptide hydrolyzing activity of the 26S proteasome comparable to that of control fibroblasts regardless

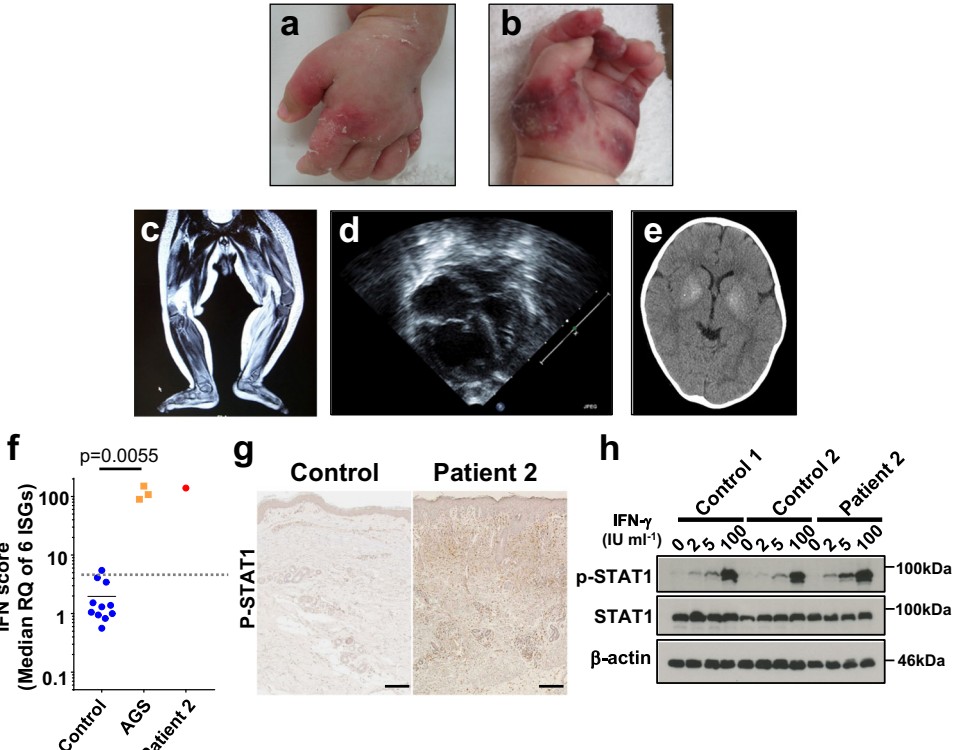

**Fig. 1 Clinical figures and IFN signature of patients. a, b** Pernio-like violaceous rash in patient 1 at 7-month-old (**a**) and patient 2 at 1-month-old (**b**). **c–e** myositis by leg magnetic resonance imaging (**c**), pulmonary hypertension by echocardiogram(**d**), and basal ganglia calcification by head computed tomography (CT) (**e**) in patient 1. **f** IFN scores of peripheral bloods of 11 healthy participants as controls, three Aicardi–Goutières syndrome (AGS) patients with IFIH1 variant and patient 2 at 84-day old. The score was regarded as positive if it exceeded $+2$ SD (5.04, dotted line) of the average (solid line) IFN score from healthy controls. Statistical analysis was performed with two-sided Mann–Whitney test. **g** The skin biopsy samples of a healthy participant as a control and patient 2 at 52-day-old stained with anti-p-STAT1 Ab. A scale bar, 100 μm. **h** The STAT1 phosphorylation assay with SV40-transformed dermal fibroblasts from healthy volunteers as controls and Patient 2. The expression levels of IFN-γ-induced p-STAT1 were enhanced in patient 2 fibroblasts compared to control fibroblasts.

**Table 1 Comparison of PRAAS, patients 1, patient 2, and *Psmb9*$^{G156D/+}$ mice.**

| | | PRAAS | Patient 1 | Patient 2 | *Psmb9*$^{G156D/+}$ mice |
|---|---|---|---|---|---|
| **Clinical findings** | Fever | + | + | + | − |
| | Skin rash | + | + | + | − |
| | Myositis | + | + | + | NA |
| | Liver dysfunction | + | + (fibrosis) | + | NA |
| | Pneumonia | + | + | + | NA |
| | Splenomegaly | + | + | + | − |
| | Lipoatrophy | + | − | − | − |
| | Pulmonary hypertension | Rare | + | + | NA |
| | Basal ganglia calcification | + | + | + | NA |
| | IFN score | Positive | − | Positive | − |
| | Viremia | − | Polyoma virus + | Polyoma virus + | NA |
| **Blood analysis** | Neutrophil | Variable | ↑ | → | ↑ |
| | Monocyte | Variable | ↑ | ↓ | ↑ |
| | NK cell | Variable | → | ↓ | → |
| | Dendritic cell | NA | NA | NA | ↓ |
| | T cell | Variable | →(CD4) ↑ (CD8) ↓ (γδ) | ↓(CD8) ↓ (γδ) | ↓ |
| | B cell | Variable | ↓ | → | ↓ |
| | Serum Ig | ↑ | ↓(IgG) | Unevaluable | ↓ |
| **Proteasome analysis** | β1i maturation | Variable | Defect | Defect | Defect |
| | Proteasome activity | 20S, 26S defect | 20S defect | 20S defect | 20S defect |
| | Ubiquitin accumulation | ++ | +/− | +/− | − |

*NA* not assessed.

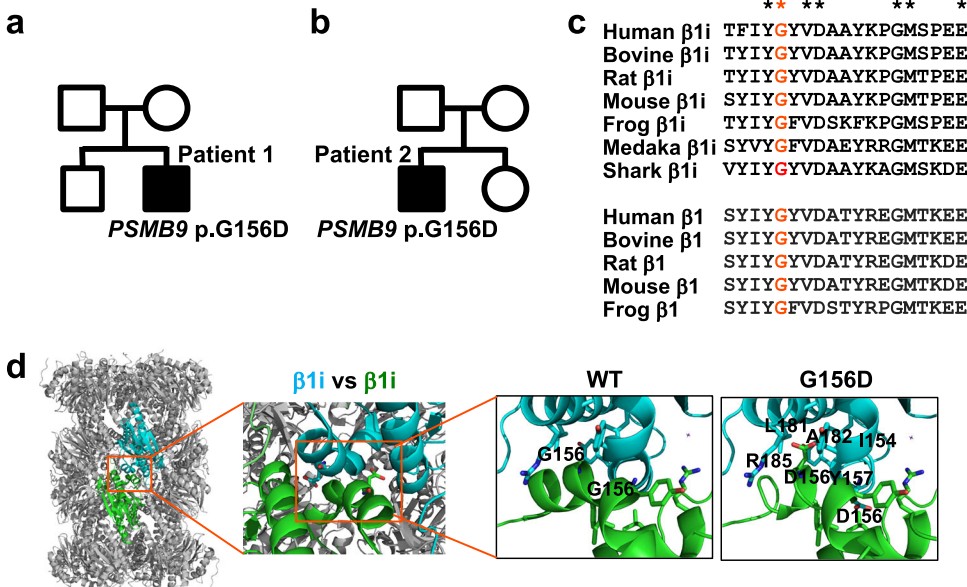

**Fig. 2 The heterozygous PSMB9 G156D variant. a, b** Pedigrees of patient 1 (**a**) and patient 2 (**b**). **c** Multiple alignments of PSMB9 (β1i) and PSMB6 (β1) in various species. Conserved amino acids among all these species are indicated by asterisks. G156 in murine PSMB9 and the corresponding amino acids are indicated in red. **d** Structures of wild type and PSMB9 G156D mutant of 20S proteasome. Structural models of PSMB9 G156D were created from the β1i-subunit structure [Protein Data Bank (PDB) ID code 3UNH]. Overall structure of 20S is shown as a ribbon model. β1i subunits are shown as green and cyan. D156 and some of the potential interacting residues of PSMB9 G156D mutant are shown in stick representation (green and cyan).

of IFN-γ-stimulation (Supplementary Fig. 4b left), whereas the activity of 20S was decreased in IFN-γ-stimulated patient's fibroblasts (Supplementary Fig. 4b right). Consistent with this, unstimulated fibroblasts showed no significant defects in components of the proteasomes (Supplementary Fig. 4c). However, IFN-γ-stimulated fibroblasts showed defects in the maturation of β1i and β5i and incorporation of β1i into the 20S and 26S proteasome and impaired catalytic activity of the 20S complex, with normal 26S proteasome activity (Supplementary Fig. 4c, d).

**Analysis of mice carrying the PSMB9 G156D mutation**. In order to clarify how the PSMB9 G156D variant contributes to the patients' manifestations, we generated mice carrying the mutation by the CRISPR/Cas9 method. $Psmb9^{G156D/G156D}$ mice died within 6–7 months old, while the survival rate of $Psmb9^{G156D/+}$ mice was similar to that of wild-type mice (Supplementary Fig. 5a). $Psmb9^{G156D/+}$ mice appeared healthy at glance and lacked a sign of lipoatrophy or inflammation (Supplementary Fig. 5b and Table 1). The $Psmb9^{G156D/+}$ and $Psmb9^{G156D/G156D}$ derived fibroblasts exhibited a peptide hydrolyzing activity of the 26S proteasome almost comparable to that of $Psmb9^{+/+}$ derived fibroblasts regardless of IFN-γ-stimulation (Supplementary Fig. 5c left), whereas the activity of 20S was decreased in IFN-γ-stimulated $Psmb9^{G156D/+}$ and $Psmb9^{G156D/G156D}$ derived fibroblasts (Supplementary Fig. 5c right). Consistent with this, the components and catalytic activities of proteasomes in unstimulated embryonic fibroblasts were comparable among $Psmb9^{+/+}$, $Psmb9^{G156D/+}$, and $Psmb9^{G156D/G156D}$ mice (Supplementary Fig. 5d, e). However, in IFN-γ-stimulated embryonic fibroblasts, immature β1i and β2i, which were undetectable in $Psmb9^{+/+}$ mice, were observed and incorporation of β1i into the 20S complex was severely impaired, although incorporation of β1i into the 26S proteasome was mildly defective in $Psmb9^{G156D/+}$ mice (Fig. 4a). Neither β1i nor β2i were hardly detected in 20S complex or the 26S proteasome of $Psmb9^{G156D/G156D}$ mice, indicating genotype-dependent defect of immunoproteasome maturation (Fig. 4a). Catalytic activity of the 20S complex was

decreased in a genotype-dependent manner (Fig. 4b). Meanwhile, in $Psmb9^{G156D/+}$ and $Psmb9^{G156D/G156D}$ mice, 26S proteasome activity was comparable to that of $Psmb9^{+/+}$ mice and accumulation of ubiquitinated proteins was unremarkable (Fig. 4b, c). Thus, similar to the patients, not only $Psmb9^{G156D/+}$ but also $Psmb9^{G156D/G156D}$ mice showed severe defects mainly in 20S complex assembly and activity.

We then performed immunological analysis. In $Psmb9^{G156D/+}$ mice, the thymus was small and the cortico-medullary junction was unclear (Fig. 5a). All thymocytes, including their subsets, were decreased in number (Supplementary Fig. 6a). Histological analysis of the spleen also showed defective formation of follicles and a decrease of B and T cells (Fig. 5b). Among splenocytes, B, as well as both CD4 and CD8 T cells, were decreased in number (Supplementary Fig. 6b). In remaining CD8 T cells, central and effector memory T cells were increased, while naïve T cells were prominently decreased in percentages (Supplementary Fig. 6c). Furthermore, serum levels of all Ig isotypes were severely decreased (Fig. 5c). Splenic dendritic cells (DCs) were also decreased in number and percentages (Fig. 5d and Supplementary Fig. 6b). Meanwhile, NK cells were normal, and neutrophils and monocytes were increased not only in spleen but also in BM (Fig. 5d, e and Supplementary Fig. 6b). In $Psmb9^{G156D/G156D}$ mice, the thymus was hardly detected, and spleen lacked T, B, NK, and DCs (Fig. 5d). Thus, $Psmb9^{G156D/+}$ mice showed combined immunodeficiency with increase of monocytes and neutrophils, and the homozygosity of this mutation caused severer immunodeficient phenotype.

In mutant mice lacking proteasome subunits, generation of MHC class I-restricted Ag peptides is impaired, which leads to decreased expression of MHC class I and decrease of CD8 T cells[13]. However, MHC class I expression was not impaired and processing and presentation of ovalbumin were also intact in splenic B cells of $Psmb9^{G156D/+}$ mice, indicating that immunoproteasome of $Psmb9^{G156D/+}$ mice functionally retained the MHC class I presentation activity (Supplementary Fig. 7).

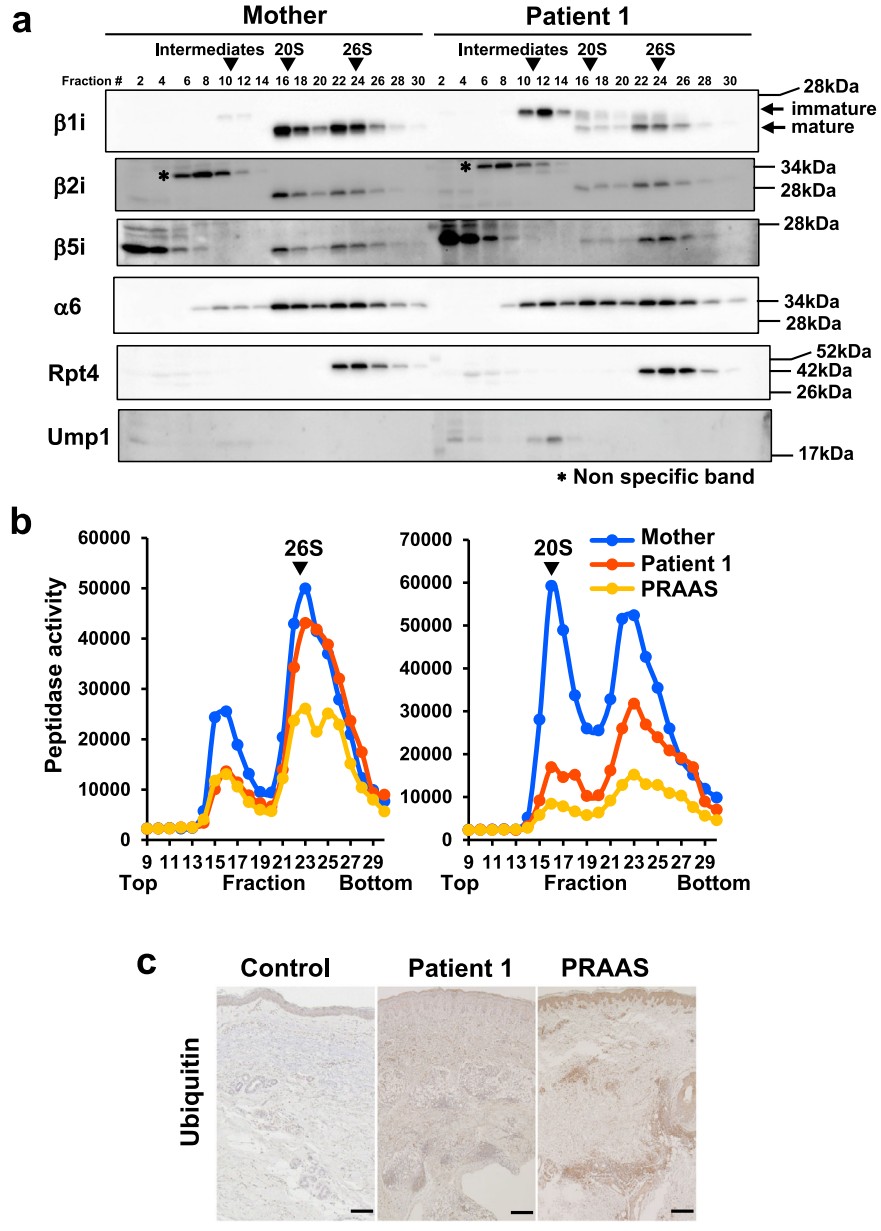

**Fig. 3 Proteasome disorder of the patient with the PSMB9 G156D variant. a, b** The proteasome analysis of immortalized B cells from patient 1 and his mother. The immortalized B cell extracts were prepared and fractionated by glycerol gradient centrifugation. Data were representative from three independent experiments. **a** Immunoblot analysis of each fraction using Abs against the indicated proteins. **b** Chymotrypsin-like activity of each fraction, which was measured by using Suc-LLVY-AMC as a substrate in the absence (left) or presence (right) of 0.0025% SDS. **c** The skin biopsy samples of a healthy control, patient 1, and a PRAAS patient stained with anti-ubiquitin Ab. Scale bars,100 μm.

## Discussion

We describe a de novo heterozygous missense variant, p.G156D, in *PSMB9* coding an immunoproteasome subunit, β1i, in two unrelated patients that manifest characteristic autoinflammation, similar to, but distinct from so far described PRAAS, with immunodeficiency. Although a heterozygous variant of *proteasome maturation protein (POMP)* leads to PRAAS, the other PRAAS-causing variants are homozygous, compound heterozygous or digenic heterozygous in proteasome subunit genes[6,8]. Noteworthy, in our patients, a monogenic heterozygous variant, *PSMB9* p.G156D, should be responsible for their manifestations.

Although the variant led to defects in β1i maturation and formation and activity of immunoproteasome, it mainly caused formation and activity defects in the 20S complex, and the 26S proteasome defects were mild enough at least to avoid ubiquitin

accumulation, as shown by the analysis not only of the present two patients but also of the mice carrying the heterozygous or homozygous PSMB9 G156D mutation (Supplementary Fig. 8). This is in contrast to published PRAAS cases, which mostly show impairment of both 20S and 26S formation and ubiquitin accumulation, accompanied with endoplasmic reticulum stress or type I interferonopathy[3,8]. According to structural modeling analysis, PSMB9 G156 is located in the interface between two β rings and G156D variant is considered to disturb the interaction of two β rings, while keeping active site conformation. Meanwhile, most of published proteasome subunit variants, as exemplified by T75M, A92T, A94P, K105Q, or G201V in PSMB8, are found at or near the active site and should affect the active site itself or its conformation (Fig. 6a), which is necessary for both 20S complex and 26S proteasome activities[5]. Thus, PSMB9 G156D is a variant in

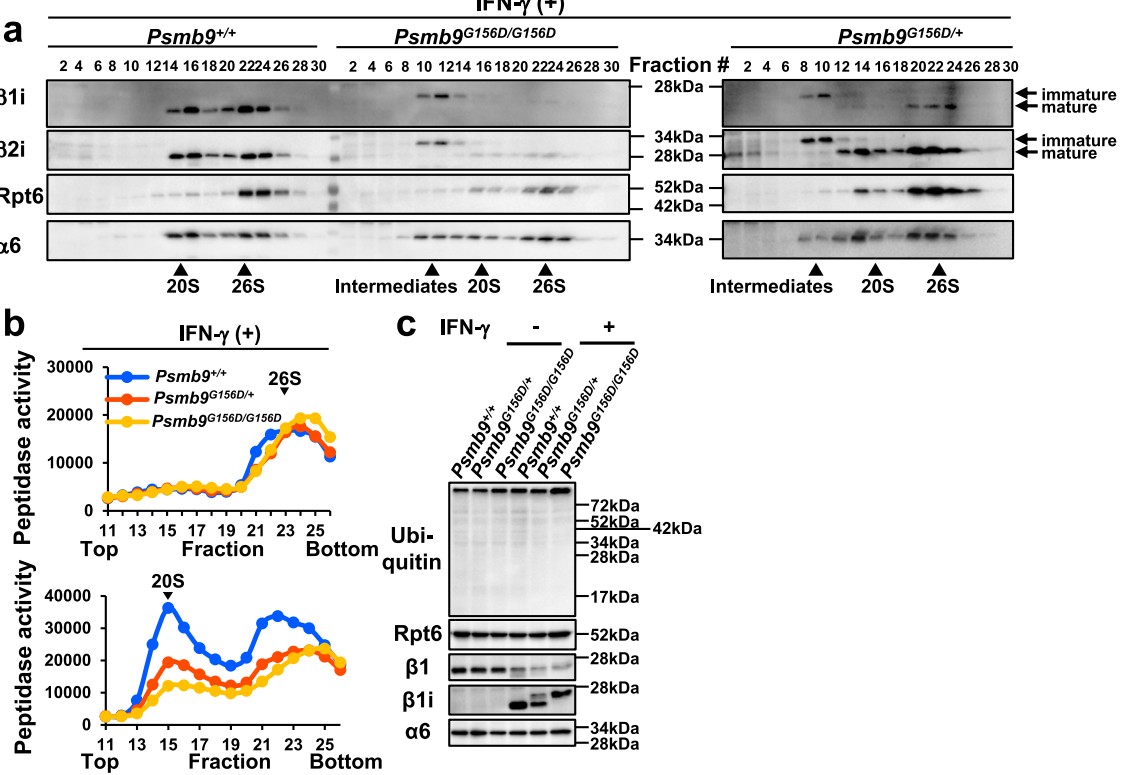

**Fig. 4 Proteasome disorder of mice carrying the PSMB9 G156D mutation. a**, **b** The proteasome analysis of IFN-γ-stimulated embryonic fibroblasts from the indicated mice. Cell extracts were prepared and fractionated by glycerol gradient centrifugation. **a** Immunoblot analysis of each fraction using Abs against the indicated proteins. Data were representative from two independent experiments. **b** Chymotrypsin-like activity of each fraction, which was measured by using Suc-LLVY-AMC as a substrate in the absence (upper) or presence (lower) of 0.0025% SDS. **c** The immunoblot analysis of embryonic fibroblast lysates using Abs against the indicated proteins. Data were representative from two independent experiments.

proteasome subunits that causes proteasome defects by its heterozygosity and affects the interaction of two β rings.

Immunodeficiency as well as proteasome defects, observed in two patients, were recapitulated in mice carrying the heterozygous PSMB9 G156D mutation. The mutant mice showed decrease of T, B cells and DCs, decrease of serum Ig levels and increase of neutrophils and monocytes. The phenotypes significantly overlap with that of patient 1, who showed decrease of B cells and serum IgG level and increase of monocytes (Fig. 5 and Table 1). Patient 2 also showed decrease of CD8 T in addition to γδ T and NK cells, although he was too young to be evaluated for B cell generation or serum IgG level. Despite the immunodeficiency, $Psmb9^{G156D/+}$ mice appeared healthy without spontaneous development of autoinflammatory phenotypes. Some environmental factors, although not defined yet, might be necessary for $Psmb9^{G156D/+}$ mice to manifest autoinflammation. Importantly, immunodeficient phenotypes in $Psmb9^{G156D/G156D}$ mice were more prominent than those in $Psmb9^{G156D/+}$ mice, suggesting the dosage effects of the PSMB9 G156D mutation. Although severe immunodeficiency should contribute to the early death of $Psmb9^{G156D/G156D}$ mice, the direct cause for the death remains unknown.

So far, mutant mice lacking either or all of β1i, β2i, β5i, and β5t have been generated to show decrease of CD8 T cell numbers and responses[13–18]. However, none of them exhibited combined immunodeficient phenotypes. Furthermore, CD8 T cell decrease observed in $Psmb9^{G156D/+}$ mice is not caused by defective immunoproteasome activity to generate MHC class I-restricted antigens, as observed in the induced immunoproteasome subunit(s)-deficient mice[13], because MHC class I expression was not

impaired and class I-restricted antigen presentation was normal in $Psmb9^{G156D/+}$ mice (Supplementary Fig. 7). It can be assumed that the variant does not lead to mere decrease, but alteration or modification of immunoproteasome functions, which cannot be monitored by catalytic properties of conventionally used substrate polypeptides. Thus, although it remains unclear at present why PSMB9 G156D leads to immunodeficiency, mice with PSMB9 G156D mutation are a faithful model for analyzing proteasome defects.

Immunodeficient phenotype is also observed in mice carrying a missense mutation of β2i, which is introduced by N-ethyl-N-nitrosourea (ENU) mutagenesis[19]. The mutation is described here as PSMB10 G209W, based on the amino acid number from the first methionine. $Psmb10^{G209W/+}$ mice showed defects of both CD4 and CD8 T cells, and $Psmb10^{G209W/G209W}$ mice lacked B as well as T cells and manifested skin disorders with hyperkeratosis and infiltration of neutrophils. G209 is highly conserved among β2 and β2i across species (Fig. 6b) and G209W mutation led to decrease of the 20S complex formation while keeping the 26S proteasome formation, which is similar to the effects of PSMB9 G156D mutation[19]. This speculation is supported by structural modeling analysis, showing that PSMB10 G209 is located at the interface of the two β rings and faced to β6 in the other half proteasome (Fig. 6a, c). Effects of PSMB10 G209W to induce immunodeficiency seem milder than those of PSMB9 G156D. Although the detailed molecular mechanisms are still unknown, it can be assumed that PSMB9 G156D and PSMB10 G209W should lead to defective interaction of two β rings and impaired formation of the 20S complex thereby causing immunodeficiencies (Supplementary Fig. 8). It also remains unknown why the 26S

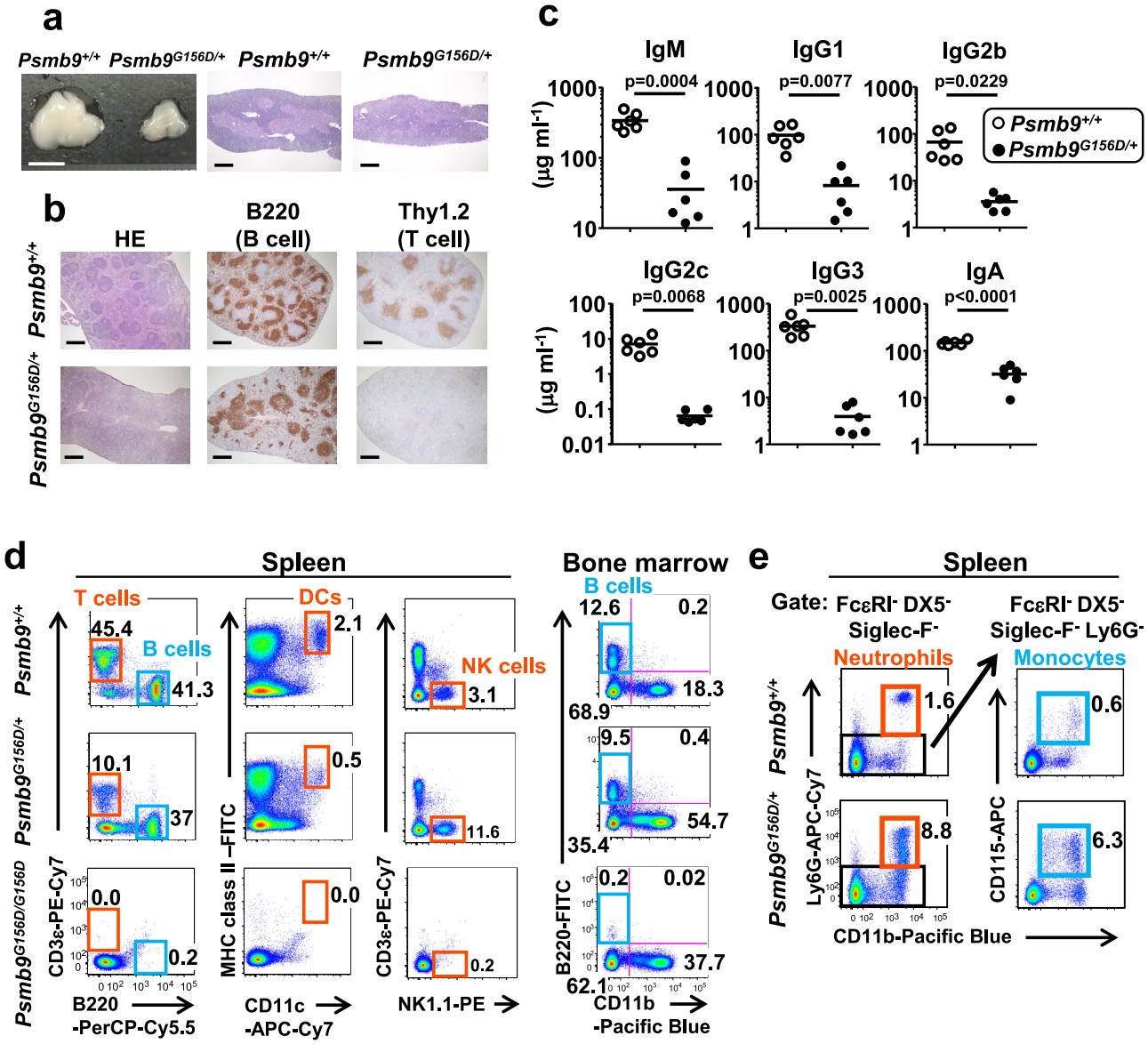

**Fig. 5 Immunological phenotype of mice carrying the PSMB9 G156D mutation. a** A photograph and histological images of the thymus. Scale bars in the leftmost and the other two panels represent 5 mm and 500 μm, respectively. Representative images from three mice were shown. **b** Cryosections of the spleen stained with hematoxylin and eosin (HE, left panel), anti-B220 Ab (middle panel), or anti-Thy1.2 Ab (right panel). Scale bars,500 μm. Representative images from three individual mice were shown. **c** Serum immunoglobulin levels. $n = 6$ in each group. Statistical analysis was performed with two-sided Welch's $t$-test. **d** FACS analysis of B and T cells, DCs, and NK cells, in the spleen or bone marrow. The numbers represent the percentages of the cells within the indicated gates or each quadrant among splenocytes or bone marrow cells. **e** FACS analysis of neutrophils and monocytes in the spleen. The left and right panels show FcεRI⁻ DX5⁻ Siglec-F⁻ and FcεRI⁻ DX5⁻ Siglec-F⁻ Ly6G⁻ cells, respectively. The numbers represent the percentages of neutrophils and monocytes among splenocytes.

proteasome formation and activity are preserved compared with the 20S complex, although 19S regulatory subunits might help facilitate the interaction of two β rings.

Ump1 coded by *POMP* is a chaperone used for proteasome assembly and heterozygous truncating variants of this gene cause autoinflammatory phenotypes with type I IFN signature[6]. The variants also result in immune dysregulation, such as lymphocyte abnormalities showing CD4 T cell increase and B cell decrease, and the authors propose a designation as POMP-related autoinflammation and immune dysregulation disease (PRAID). The manifestations are similar, but distinct from those of our cases (Table 1) and *Psmb9^{G156D/+}* mice, because the POMP variants lead to both the 20S complex and the 26S proteasome defects with

ubiquitin accumulation and hyper-γ-globulinemia with autoantibodies[6].

Here we would propose a designation "proteasome-associated autoinflammatory syndrome with immunodeficiency (PRAAS-ID)" to indicate a category of autoinflammatory diseases, similar to, but distinct from PRAAS, as represented by the patients in this study. PSMB9 G156D mutant mice are useful to clarify not only the pathogenesis of proteasome dysfunctions but also homeostatic roles of immunoproteasome and to develop effective therapeutic maneuvers for PRAAS and immunodeficiency. It is also noteworthy that proteasome-related genes be candidate genes for screening patients with primary immunodeficiency of unknown etiology.

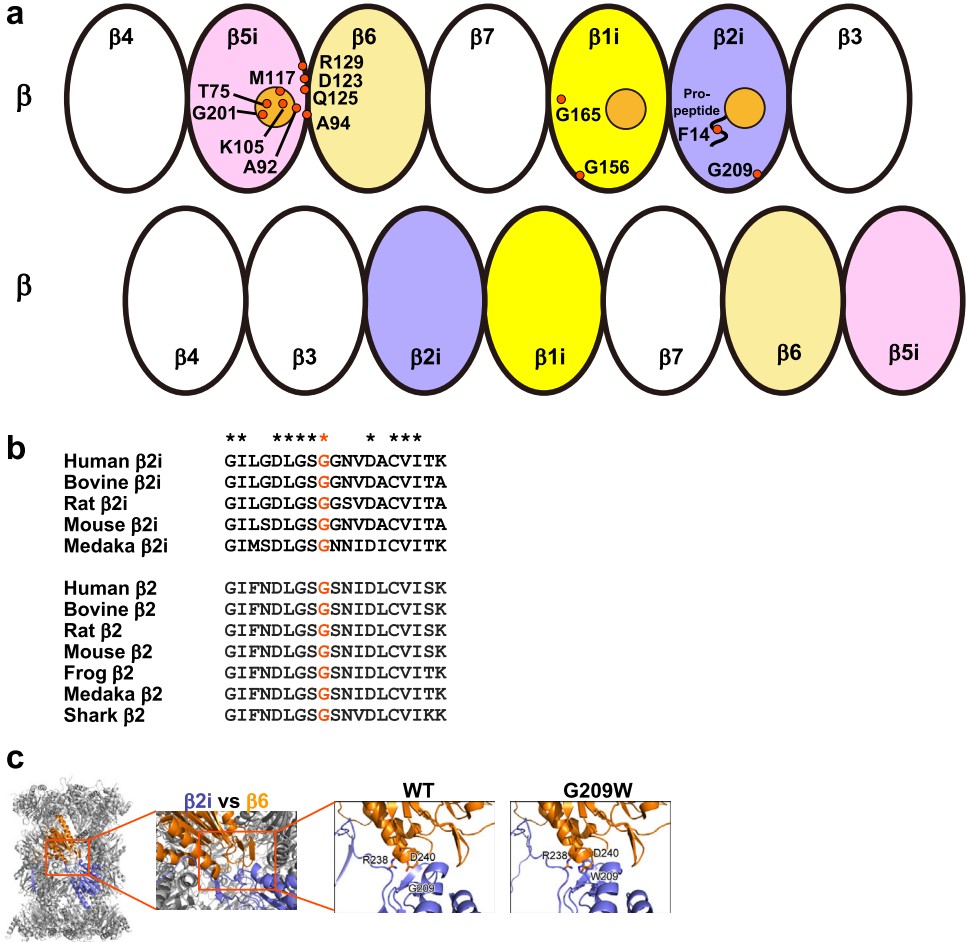

**Fig. 6 Characterization of proteasome subunit missense variants. a** Locational distribution of published missense variant sites on the schematic model of β rings in the immunoproteasome (https://infevers.umai-montpellier.fr/web/index.php). The mutant residues are represented by red circles. Catalytic active sites in β1i, β2i, and β5i are marked with orange circles. Notably, PSMB9 G156 and PSMB10 G209 are at the interface of two β rings, while most of mutated residues in PSMB8 are at or around active sites. **b** Multiple alignments of PSMB10 (β2i) and PSMB2 (β2) in various species. Conserved amino acids among all these species are indicated by asterisks. G209 in murine PSMB10 and the corresponding amino acids are indicated in red. **c** The structure model of wildtype and PSMB10 G209W mutant of the 20S complex, based on the β2i-subunit structure [Protein Data Bank (PDB) ID code 3UNH]. The overall structure of the 20S complex is shown as a ribbon model. β2i and β6 subunits are shown as light blue and orange, respectively. W209 and some of the potential interacting residues of PSMB10 G209W mutant are shown in stick representation.

## Methods

**Patients**. Written informed consent was obtained from the patients' parents and all other study participants.

**Approval for this study**. Studies on human participants were approved by the Ethics Committee of the University of the Ryukyus for Medical and Health Research Involving Human Subjects, the ethics boards of Gifu University, the Ethics Committee on Human Genome / Gene Analysis Research Nagasaki University, and the Research Ethics Committee of Wakayama Medical University. All animal experiments were done at Wakayama Medical University and approved by the Animal Research Committee of Wakayama Medical University.

**Genetic studies**. Whole exome sequencing was performed by using genomic DNAs obtained from peripheral blood of patients and parents[20]. In brief, exon fragments were enriched from genomic DNA samples of a patient and his parents using the SureSelect Human All Exon Kit V5 kit (Agilent Technologies), according to the manufacturer's instructions. The prepared libraries were sequenced by the HiSeq2500 sequencer (Illumina) to obtain 100 bp+100 bp paired-end reads. The reads were mapped by the Novoalign software (Novocraft Technologies Sdn Bhd) on the hg19 human reference genome. The Genome Analysis Toolkit was used for following local realignment and variants call to obtain SNV and small insertion/deletion (indel) calls combined with in-house workflow management tool[21,22]. Using data from the patient and his parents, called variants that can be de novo variants in the patients were selected using an in-house tool. Selected variants were annotated by using the ANNOVAR software[23]. Finally, the variants that meet the following criteria were selected as "deleterious" variants: (1) leads stop gain, stop loss, nonsynonymous variant, or splice site variant, (2) alternative allele frequencies in databases including Complete Genomics whole genome data, the 1000 genome project, NHLBI GO ESP, the human genome variation database of the Japanese population[24] and an in-house database are all ≤0.5%, (3) not included in segmental duplication region defined in the UCSC genome browser[25,26].

For target capture sequencing, ubiquitin-proteasome-system, autophagy and interferonopathy-related gene panel were purchased from Integrated DNA Technology (IDT) (Coralville, IA). Gene capture and library construction were performed using Hybridization capture of DNA libraries using xGen Lockdown Probes and Reagents (IDT) according to the manufacturer's instructions. The libraries were sequenced by the MiSeq sequencer to obtain 300 bp+300 bp paired-end reads. For the reads, mapping, SNV/indel calling, annotation, and narrowing were performed as with the exome analysis. Identified candidate variants were confirmed by Sanger sequencing. *NEMO* gene was amplified with long-range PCR and then the regions of exon and exon-intron boundaries were amplified by PCR for direct sequencing[27].

TREC and KREC levels in the patients' peripheral blood mononuclear cells (PBMC) were measured by real-time polymerase chain reaction (PCR). Genomic DNA from peripheral blood was prepared, and TREC and KREC levels were measured by real-time PCR (LightCycler 96, Roche Diagnostics). Primer pairs and probes are shown in Supplementary Table 5. A range of more or <100 copies/ug DNA is defined as positive or undetectable, respectively[28].

**IFN score**. Total RNA was extracted from human blood samples collected into PAXgene Blood RNA tubes (762165, Becton, Dickinson and Company) and was

reverse transcribed by PrimeScript II 1ˢᵗ strand cDNA synthesis kit (6210 A, Takara). Quantitative real-time PCR was performed with Taqman Gene Expression Master Mix (4369016, Applied Biosystems) and probes (Supplementary Table 5) on StepOnePlus Real-Time PCR system (Applied Biosystems). The expression levels of each transcript were determined in triplicate and normalized to the level of β-actin. Results were shown relative to a single calibrator. Median fold change of expression levels of the six IFN-stimulated genes was used to calculate the IFN score for each patient. The score was regarded as positive if it exceeded +2 SD of the average IFN score from healthy controls[29,30].

To calculate IFN score in mice, total RNA was extracted with RNeasy micro kit (QIAGEN) from $Psmb9^{+/+}$ and $Psmb9^{G156D/+}$ splenocytes and reverse transcribed by PrimeScript RT reagent Kit (RR037A, Takara). Quantitative real-time PCR was performed with TB Green Premix Ex Taq II (RR820A, Takara) and specific primers for six IFN-stimulated genes on the StepOnePlus Real-Time PCR system (Applied Biosystems). The primers are shown in Supplementary Table 5. The expression levels of each transcript were normalized to the levels of 18S ribosomal RNA, which were determined by TaqMan probes (TaqMan Gene Expression Assay, Applied Biosystems). Median fold change of the six IFN-stimulated genes was used to calculate the IFN score for $Psmb9^{+/+}$ and $Psmb9^{G156D/+}$ mice. The score was regarded as positive if it exceeded +2 SD of the average IFN score from $Psmb9^{+/+}$ mice.

**Predictive analysis of the protein structure.** Structural models of wildtype and mutated immunoproteasomes were created based on the structure of the immunoproteasome [Protein Data Bank (PDB) ID code 3UNH] using the Swiss-Model server[31] and CNS (Crystallography and NMR system) program[32].

**Cytokine measurement by ELISA.** Human serum samples were stored at −80 °C until assayed. The concentrations of TNF-α, IL- 1β, IL-6, IL-18, IFN-α, and IP-10 were measured with ELISA kits (Invitrogen for TNF-α, IL-1β, and IL-6, R&D for IP-10, MBL for IL-18, and PBL Assay Science for IFN-α).

**Western blot analysis of human fibroblasts.** SV40 transformed fibroblasts were stimulated with 1000 IU mL⁻¹ of IFN-γ for 15 min. The total cell lysates were subjected to SDS-PAGE. Expression of p-STAT1, STAT1, and β-actin was evaluated with anti-p-STAT1 (pY701) Ab (58D6, Cell Signaling 9167, 1:1000), rabbit polyclonal anti-STAT1 Abs (Stat1α (C-24), SANTA CRUZ sc-345, 1:1000) and anti-β-actin Ab (AC-74, Sigma-Aldrich A2228, AC-74, 1:2000), respectively.

**Immunohistochemical analysis of human samples.** Six micrometer sections of skin biopsy samples from healthy controls and patients were stained with rabbit polyclonal anti-p-STAT1 Abs (GeneTex GTX50118, 1:590) and anti-ubiquitin Abs (DAKO Z0458, 1:3000).

**Generation of the mutant mice.** $Psmb9$ G156D mutant mice were generated by the CRISPR/Cas9 method. Briefly, guide RNA (gRNA) targeting to $Psmb9$ exon5 (guide sequence; 5'- CTCCTACATTTAGGTTATG -3') and mRNA for Cas9 endonuclease were generated by in vitro transcription using MEGAshortscript T7 (Life Technologies) and mMESSAGE mMACHINE T7 ULTRA kit (Life Technologies), respectively. The synthesized gRNA and Cas9 mRNA were purified with MEGAclear kit (Life Technologies). A single-stranded oligodeoxynucleotides (ssODN) containing the c.G467A mutation (5'- CATCTGTGGTGAAACGCCGGC ACTCCTCAGGGGTCATGCTGGCTTATAAGCTGCGTCCACATAATCATA AATGTAGGAGCTTCCGGAACCGCCGATGGTAAAGGGCTGTCGAATTAG CATCCCTCCCATG -3') was synthesized by Integrated DNA Technologies. To generate mutant mice, female B6C3F1 mice were superovulated and mated with C57BL/6 N males (Clea Japan). Fertilized one-cell-stage embryos were injected with gRNA, Cas9 mRNA, and ssODN. The c.G467A mutation was confirmed by sanger sequencing, and the mutant mice were further backcrossed to C57BL/6 N mice for more than six generations. Eight to sixteen weeks old mutant mice and their littermates were analyzed.

In order to generate OVA-expressing $Psmb9$ G156D mice, $Psmb9$ G156D mutant mice were crossed with membrane-bound OVA-expressing transgenic mice (C57BL/6-Tg(CAG-OVA)916Jen/J, the Jackson Laboratory, stock number 005145)[33].

All mice were housed under specific pathogen-free conditions in a 12 h/12 h light/dark cycles at an ambient temperature of 20–24 °C and a humidity range 40–60%. Experimental/control animals were co-housed and used at 8–16 weeks old. Both males and females were used in the study. Mice were euthanized by cervical dislocation before dissection of tissues. All animal experiments were conducted in accordance with the Guidelines for the Care and Use of Laboratory Animals on Wakayama Medical University and with the relevant national guidelines and regulations.

**Cell preparation.** Immortalized B cells were generated by infecting patient 1-derived peripheral blood cells with Epstein–Barr virus. Fibroblasts were generated from patient 2-derived skin sample. The control human fibroblasts were

purchased from TOYOBO, Osaka, Japan:106–05a; and KURABO, Osaka, Japan: KF-4109. The fibroblasts were transformed with origin-defective mutant of SV40 virus carrying wild-type T antigen.

For preparing mouse embryonic fibroblasts, E13.5–14.5 embryos were dissected, cut into small pieces, and soaked in 0.25% trypsin/EDTA (Nacalai Tesque) at 37 °C with shaking for 30 min. The cells were then suspended by pipetting, filtrated with 100 μm pore nylon mesh, and plated on a 15 cm dish per embryo and cultured in DMEM supplemented with 10% FBS. Cells cultured during days 5–10 were used for experiments.

**Flow cytometry.** For the analysis of human lymphocyte subsets, peripheral blood samples were stained with fluorochrome-conjugated or biotinylated Abs (Supplementary Table 6). Cells were analyzed on a Navios EX flow cytometer with Kaluza version 2.1 (Beckman Coulter).

Single-cell suspensions of splenocytes and bone marrow cells from $Psmb9^{+/+}$, $Psmb9^{G156D/+}$, and $Psmb9^{G156D/G156D}$ were incubated with an Ab against CD16/32 to block non-specific binding of Abs to Fc receptors. Then, the cells were stained with fluorochrome-conjugated or biotinylated Abs (Supplementary Table 6). Dead cells were excluded by staining with Fixable Viability Dye (eBioscience). To detect MHC class I-mediated presentation of endogenous OVA, splenocytes were stained with an Ab against OVA-derived peptide/H2-Kᵇ complex (Supplementary Table 6). Cells were analyzed on a FACS Verse with FACSuite software v1.0.5.3841 or FACS Aria II with FACSDiva software version 8.0.2 (BD Biosciences) and data were analyzed with FlowJo software version 9.9.6 (BD Bioscience). Representative gating strategies are shown in Supplementary Fig. 9.

**Cell lysates, glycerol gradient analysis, measurement of proteasomal activity, and western blotting.** Cells were lysed in buffer containing 25 mM Tris-HCl (pH7.5), 0.2% NP-40, 1 mM DTT, 2 mM ATP, 5 mM MgCl₂, and 1 mM phenylmethylsulfonyl fluoride (PMSF) and clarified by centrifugation at 20,000 × g for 10 min at 4 °C. For glycerol gradient centrifugation analysis, clarified cell lysates were subjected to 8–32% (v/v) glycerol linear density gradient centrifugation (22 h, 83,000 × g) and separated into 32 fractions, and peptidase activities of each fraction was measured using a fluorescent peptide substrate, succinyl-Leu-Leu-VI-Tyr-7-amido-4-methylcoumarin (Suc-LLVY-AMC), for chymotrypsin-like activity with ARVO MX 1420 (Perkin Elmer)[34]. To verify that the peptidase activity was dependent on the proteasome, cells were treated with a selective proteasome inhibitor, carfilzomib (CFZ, LC laboratories), at 100 nM for 6 hr. The Abs against Ump1, β1i, β2i, β5i, β1, β2, β5, α2, α6, and polyubiquitin were established by S. Murata[34–37].

**Histology in mice.** Thymus and spleen were fixed with 4% paraformaldehyde and embedded in OCT compound (Sakura Finetek) or FSC22 frozen section compound (Leica Microsystems), and 5 μm cryosections were prepared. The sections were stained with hematoxylin and eosin (Muto Pure Chemicals). To detect B and T cells, sections were stained with biotinylated Abs against B220 (RA3-6B2, eBioscience 13-0452-85, 1 ug mL⁻¹) or Thy1.2 (30-H12, BD Bioscience 553011, 1 ug mL⁻¹) after blockade of endogenous peroxidase with 3% H₂O₂/methanol. Biotinylated Abs were developed with VECTASTAIN ABC Standard kit and ImmPACT DAB (Vector Laboratories). Hematoxylin was used as counterstain.

**Measurement of serum Ig levels in mice.** Serum Ig levels were measured by ELISA developed in hand. In brief, serum samples were incubated with ELISA plate coated with goat anti-mouse IgM, IgG1, IgG2b, IgG2c, IgG3, or IgA Abs (Southern Biotech) and detected with biotinylated goat Abs against each class (Southern Biotech) and streptavidin-conjugated alkaline phosphatase. The plate was developed by alkaline phosphatase buffer (50 mM NaHCO₃, 10 mM MgCl₂, pH9.8) containing phosphatase substrate tablet (S0942, Sigma-Aldrich). Purified mouse IgM, IgG1, IgG2b, IgG2c, IgG3 (Southern Biotech) were used as standard.

**Statistical analysis.** Statistical significance was determined using two-sided Student's t-test, two-sided Welch's t-test, two-sided Mann–Whitney test, or Log-rank test as indicated in the figure legends using GraphPad Prism 7 or 9 (GraphPad Software). $p < 0.05$ was considered significant.

**Reporting summary.** Further information on research design is available in the Nature Research Reporting Summary linked to this article.

## Data availability

The exome sequencing datasets generated and analyzed in this study are not publicly available due to patient privacy and confidentiality but are available from the corresponding author on reasonable request. The sequence data on $PSMB9$ c.467 G > A (p.G156D) variant is deposited in NCBI ClinVar under the accession number VCV001299361 https://www.ncbi.nlm.nih.gov/clinvar/variation/1299361/. Source data are provided with this paper.

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

## Acknowledgements

We thank Dr. Masahiro Yamamoto for useful advice and discussion for the CRISPR/Cas9 method, Ms. Ikuko Hattori and Ms. Chihiro Nakai for technical assistance, Ms. Aoi Tawaki-Matsumura and Ms. Akane Nishiwaki for secretarial assistance and Dr. Iwao Nakazato for preparation of histological results. This work was supported by Health Labor Sciences Research Grants for Research on Intractable Diseases (grant 20316700 and 20317089 to H.O.) from the Ministry of Health, Labor and Welfare (MHLW) of Japan; by the Practical Research Project for Rare/Intractable Diseases under Grant Numbers JP15gk0110012 (to K.-i.Y.), JP16ek0109179 and JP19ek0109209 (to S.O.), JP17ek0109100 (to N. Kan), JP19ek0109199 (to, N. Kan, H.H., N. Ki, K.-i.Y., and T. Kai), JP20ek0109480 (to H.O. and S.O.), and Advanced Research and Development Programs for Medical Innovation (AMED-CREST) under Grant Number JP20gm1110003 (to S.M.) from the Japan Agency for Medical Research and Development (AMED); by Grants-in-Aids for Scientific Research (A) (grant JP18H04022 to S.M.), Scientific Research (B) (grant JP16H05355 and JP19H03620 to S.O., grant JP17H04088 and JP20H03505 to T. Kai), for Scientific Research (C) (grant JP15K09780 and JP19K08798 to N. Kan, grant JP18K07840 to H.O., grant JP19K07628 to I.S., grant JP18K07071 to H.H., grant JP16K10171 and JP19K08754 to K.K.), for Young Scientists (grant JP19K17293 to K.I., grant JP20K16289 to T. Or, JP18K16096 to Y.F.-O.), for Scientific Research on Innovative Areas (grant JP17H05799 and JP19H04813 to T. Kai, JP18H05500 to S.M.), for Exploratory Research (grant JP17K19568 to T. Kai), for Young Scientists (B) (grant JP16K19585 to Y.F.-O.), Research Activity start-up (grant JP19K23848 to T. Or), Promotion of Joint International Research from the Japan Society for the Promotion of Science (JP18KK0228 to S.O.) from the Japan Society for the Promotion of Science; by the Uehara Memorial Foundation (to I.S. and T. Kai); by Takeda Science Foundation (to H.H., T.M., I.S., and T. Kai); by the Ichiro Kanehara Foundation for the promotion of Medical Sciences and Medical care (to H.H.); by the Inamori Foundation (to I.S.); by the Extramural Collaborative Research Grant of Cancer Research Institute, Kanazawa University; by a Cooperative Research Grant from the Institute for Enzyme Research, Joint Usage/Research Center, Tokushima University; by the Grant for Joint Research Program of the Institute for Genetic Medicine Hokkaido University; by the Grant for Joint Research Project of the Institute of Medical Science, the University of Tokyo; by the Program of the Network-type Joint Usage/Research Center for Radiation Disaster Medical Science; by Wakayama Medical University Special Grant-in-Aid for Research Projects.

## Author contributions

N. Kanazawa, H.H., H.O. and T. Kai designed the research. N. Kanazawa, J.H., Y.H., K.I., R.N., M.T., Y.Y., S.T., K.K., S.O., T.T. and S.M. performed biochemical and histochemical experiments. H.H., T. Or, T. Oz, T. K., I.S., Y.F.-O., N.W.-N., Y.I. and T. Kai generated and analyzed a murine model. N. Kinjo, H.O., S.H., K.H., N. Kawamoto, S.K. and K.N. saw the patients and analyzed the patients' materials. H.M., A.K. and K.-i.Y. performed genomic analyses. T.M. performed simulation analyses. N. Kanazawa, H.H., N. Kinjo, H.O., J.H., H.M., S.M., K.-i.Y. and T. Kai wrote and edited the manuscript.

## Competing interests

The authors declare no competing interests.
