## [Peer Review File · Nature Communications]

Heterozygous missense variant of the proteasome subunit β -type 9 causes neonatal-onset autoinflammation and immunodeficiencyREVIEWER COMMENTS

Reviewer #1 (Remarks to the Author):

Very interesting paper

Could the authors explain why they definitively exclude possible digenism, and discuss more Brehm et al?

Reviewer #2 (Remarks to the Author):

In their study entitled 'Neonatal-onset autoinflammation and immunodeficiency caused by heterozygous missense mutation of the proteasome subunit beta t-type 9' Kanazawa et al. report two unrelated patients that carry a heterozygous G156D mutation in the immunoproteasome subunit PSMB9 (beta1i/LMP2). This mutation lies not at the active site substrate binding pocket but at a position where two subunits of the two beta rings of the 20S immunoproteasome face each other. They show that the maturation of immunoproteasomes is impaired in cells from these patients and in cells from a PSMB9-G156D mutant mouse which they have generated. Chymotrypsin-like activity of the 20S but curiously not the 26S proteasome is impaired and immunoproteasome assembly intermediates accumulate. Consistently, an accumulation of ubiquitin conjugates has not been observed. The patients show rash, pulmonary hypertension, and basal ganglia calcification. They also show signs of autoimmune myositis. The mutant mice show reduced numbers of thymocytes and splenocytes with less T cells and B cells but more neutrophils and monocytes. The MHC class I expression is not affected. Taken together, a new disease-related immunoproteasome mutation has been described in humans and has been verified as the cause of disease symptoms in mice. The proteasome assembly defects, immunodeficiency, autoinflammation and disease symptoms resemble in many ways a recently reported PSMB10-G209W mutation in mice (Ref. 18). While the molecular and clinical characterizations of the new G156D mutation are interesting we do not learn much new about immunoproteasome function or assembly in mechanistic terms and numerous PRAAS causing mutations in immunoproteasome genes have been described in humans. The study may therefore be more suitable for a clinical journal.

The authors propose that this disease shall be subsumed with other PRAAS diseases under a new umbrella name 'proteasome-associated autoinflammation and immunodeficiency disease (PRAID)' because the PSMB9-G156 mutation causes an autoimmune disease (autoimmune myositis) which other autoinflammatory PRAAS disease do not. I do not think that this new nomenclature is justified because POMP1 mutant humans also show autoantibodies (Ref. 6) and that autoimmunity contributes to inflammatory symptoms in other PRAAS families has not been excluded.

The experimental work on the effect of the G156D mutation on the immunoproteasome assembly and function as well as the effect on the immune system is solid. The clinical documentation I can't judge so well but it seems to be adequate as well. The article is well written. I have only a few requests for control experiments and minor text corrections:

Fig. 1g: a comparison to a skin biopsy from a healthy control should be provided

Fig. 3: please indicate in the legend from what kind of cells the extract has been prepared. A negative control for the ubiquitin staining in Fig. 3C should be displayed.

Extended Data Fig. 1: a negative control for anti-phospho-STAT1 staining from a healthy control should be displayed in panel 1f and for the H&E staining of muscle in panel 1b a healthy control would be needed as well.

Extended Data Fig. 4: in panel 4a a healthy control is missing for the anti-ubiquitin staining. In panel b the chymotrypsin-like activity of the proteasome is measured in total lysate using the fluorogenic substrate Suc-LLVY-AMC. It has been shown that this substrate can be cleaved by cellular proteases other than the proteasome. Therefore, the peptidase activity at least for the

controls should be measured in the absence and presence of a highly selective proteasome inhibitor like lactacystin or epoxomicin.

Extended Data Fig. 5c: It has been shown that Suc-LLVY-AMC can be cleaved by cellular proteases other than the proteasome. Therefore, the peptidase activity at least for the controls should be measured in the absence and presence of a highly selective proteasome inhibitor like lactacystin or epoxomicin.

Introduction page 4: the definite article 'the' is needed in front of the words: Proteasome (line 7), 20S (line 9), immunoproteasome (line 13), thymoproteasome (line 14).

Reviewer #3 (Remarks to the Author):

General comments:

The manuscript by Kanazawa N et al reports two unrelated patients with the same de novo missense variant in PSMB9. Both patients presented with clinical manifestations observed in previously reported CANDLE/PRAAS patients in addition to immunodeficiency. Proteasome protease activity dysfunction was detected in patients' cells. The authors generated a mouse model harboring the patients' variant in which the proteasome defect and immunodeficiency were recapitulated, although the mice did not have a clinical phenotype. The manuscript clearly depicted proteasome function assays both in human and mouse cells.

I have the following comments below:

- Throughout the text, ideally "mutation" should be replaced with "variant" to be in accordance with recent genetic nomenclature.

Abstract:

"We propose the term, proteasome-associated autoinflammation and immunodeficiency disease (PRAID), as an umbrella name for our cases, PRAAS with immunodeficiency, as well as PRAAS described so far."

The sentence is not clear to me. Are the authors suggesting that PRAAS and PRAAS with immunodeficiency (ID) be a subtype of PRAID or that only the described cases of PRAAS with ID be called PRAID? In addition, as the authors also mention in the discussion, PRAID has been proposed by Poli et al as a term for "POMP-related autoinflammation and immune dysregulation disease".

Results:

The authors start describing patient 2 instead of patient 1. I understand patient 2 had evidence of an IFN signature whereas patient 1 did not have but it would be better to start the clinical part of the results section with patient 1 for clarity.

I also have the following specific comments:

Page 4 Line 25: "These patients showed the common features, such as..." I think this statement is incomplete. Did you mean "These patients showed the common features of PRAAS, such as..."?

Page 4 Line 31: Please replace "central spinal fluid" with "cerebrospinal fluid".

Page 4 Line 34: "Patient 2 also showed manifestations compatible with hemophagocytic lymphohistiocytosis."

Page 5 Lines 4-5: "Interestingly, polyoma (BK/JC) viruses were detected in both patient's peripheral bloods."

- What treatment were the patients receiving or at what stage of their disease they were when BK and JC viruses were detected? It would be good to have a brief description of both patients' therapies in the main manuscript in addition to the supplementary appendix.
- Replace "bloods" with "blood" or with "blood samples"

Page 5 Lines 9-12: The authors mention that the presence of pulmonary hypertension present in their patients is a difference between them and PRAAS patients. However, in Sanchez et al, JCI, 2018, one of the 10 CANDLE/PRAAS patients reported had pulmonary hypertension. Regarding the presence of immunodeficiency, the two PRAAS2 (or PRAID) patients reported by Poli et al, AJHG, 2019, also presented with immunodeficiency.

Page 5 Lines 14-30: The presence of the same de novo variants in PSMB9 in the 2 unrelated patients and the functional assays indicate that PSMB9 is the best candidate gene in the patients. However, there are many similarities between the patients herein reported and patients with NEMO-deleted exon 5 autoinflammatory syndrome (NEMO-NDAS) described by de Jesus et al, JCI, 2020. In NEMO-NDAS patients, the IKBKG variants are not detected by standard WES analysis and bam files should be manually inspected for the region where these variants were reported. I think it would be important to rule out NEMO-NDAS for the 2 patients reported.

Page 5 Lines 20-22: "The latter, a rare polymorphism (rs183923514), was also detected in his father without any autoinflammatory manifestations (Fig. 2a) and 11G is not conserved even in mouse."

Variant PSMD9 p.G11S has a minor allele frequency (MAF) of 0.0004483 (gnomAD) being more frequent in the East Asian population (0.005745). The variant is rare but is not a polymorphism as the MAF is below 1%. Moreover, since PRAAS can have a digenic inheritance mode (Brehm et al, JCI, 2015), the fact that the unaffected father carries the variant does not rule out it as candidate. I do agree that the PSMD9 most likely is not causing disease in the patient (based on conservation and MAF in East Asians) but the sentence should be rephrased.

Page 5 Line 36: "... patient 1-derived cells..." What cells? Please specify the type of cell here.

Discussion

Page 9 Lines 6-13: "Here we would propose a new designation "proteasome-associated autoinflammation and immunodeficiency disease (PRAID)" as an umbrella name for our cases as well as PRAAS and POMP-RAID"

Since Poli et al suggested PRAID as a term for "POMP- related autoinflammation and immune dysregulation disease", I'm not sure PRAID could be used for "proteasome-associated autoinflammation and immunodeficiency disease". The authors may want to consider PRAASID for "PRAAS with immune dysregulation".

Table 1: For pulmonary hypertension, PRAAS should be "rare" but not absent as Sanchez et al, JCI, 2018, reported the finding in 1/10 PRAAS patients.

Extended Data Fig. 1: I understand the figure shows the clinical and laboratory findings of patient 1 with a normal IFN score and negative pSTAT1 staining in patient's skin biopsy, differently from patient 2 (Fig. 1). It's not clear in the manuscript text that patient 1 did not have evidence of an IFN signature. It would be important to state in which context the IFN score was measured and the skin biopsy was performed. Was IP-10 measured for patient 1?

We thank the reviewers and editors for the critical and positive comments to improve our manuscript. In the followings, we describe point-by-point responses to the reviewer's comments.

Reviewer #1 (Remarks to the Author):

Very interesting paper

Could the authors explain why they definitively exclude possible digenism, and discuss more Brehm et al?

Through the re-evaluation, we found that the allele frequency of *PSMD9* (NM_002813) c.31G>A (p.G11S) variant in patient 1 was 0.0135 in Japanese population (Tohoku University Medical Megabank Organization: ToMMO). This indicates that pathogenicity of this variant is very unlikely. Furthermore, no other candidate variants among the proteasome-related genes except *PSMB9* p.G156D variant were found in patient 2. Thus, we assume that manifestations of the patients in this study are caused by a monogenic heterozygous variant of a proteasome-related gene, although possibility of digenism is not definitively excluded. This was described in page 7. Furthermore, the part, 'The latter, a rare polymorphism (rs183923514), was also detected in his father without any autoinflammatory manifestations (Fig. 2a) and 11G is not conserved even in mouse.' (page5 lines 20-22 of the revised manuscript) was removed. '*PSMD9* p.G11S' was also removed from Fig.2a.

Reviewer #2 (Remarks to the Author):

In their study entitled 'Neonatal-onset autoinflammation and immunodeficiency caused by heterozygous missense mutation of the proteasome subunit beta t-type 9'; Kanazawa et al. report two unrelated patients that carry a heterozygous G156D mutation in the immunoproteasome subunit PSMB9 (beta1i/LMP2). This mutation lies not at the active site substrate binding pocket but at a position where two subunits of the two beta rings of the 20S immunoproteasome face each other. They show that the maturation of immunoproteasomes is impaired in cells from these patients and in cells from a PSMB9-G156D mutant mouse which they have generated. Chymotrypsin-like activity of the 20S but curiously not the 26S proteasome is impaired and immunoproteasome assembly intermediates accumulate. Consistently, an accumulation of ubiquitin conjugates has not been observed. The patients show rash, pulmonary hypertension, and basal ganglia calcification. They also show signs of autoimmune myositis. The mutant mice show reduced numbers of thymocytes and splenocytes with less T cells and B cells but more neutrophils and monocytes. The MHC class I expression is not affected. Taken together, a new disease-related immunoproteasome mutation has been described in humans and has been verified as the cause of disease symptoms in mice. The proteasome assembly defects, immunodeficiency, autoinflammation and disease symptoms resemble in many ways a recently reported PSMB10-G209W mutation in mice (Ref. 18). While the molecular

and clinical characterizations of the new G156D mutation are interesting we do not learn much new about immunoproteasome function or assembly in mechanistic terms and numerous PRAAS causing mutations in immunoproteasome genes have been described in humans. The study may therefore be more suitable for a clinical journal.

We appreciate this critical comment. We here analyzed not only patients but also mutant mice carrying the novel *PSMB9* variant and firmly believe that overall results and conclusion should be interesting enough to a broad range of readers for Nature Communications.

The authors propose that this disease shall be subsumed with other PRAAS diseases under a new umbrella name 'proteasome-associated autoinflammation and immunodeficiency disease (PRAID)' because the PSMB9-G156 mutation causes an autoimmune disease (autoimmune myositis) which other autoinflammatory PRAAS disease do not. I do not think that this new nomenclature is justified because POMPI mutant humans also show autoantibodies (Ref. 6) and that autoimmunity contributes to inflammatory symptoms in other PRAAS families has not been excluded.

POMP variant cases are positive in various autoantibodies and show immune dysregulation rather than immunodeficiency (Poli et al 2018 Am J Hum Genet). Meanwhile, two patients carrying *PSMB9* G156D variant in this study showed their manifestations at neonatal-onset and were negative for autoantibodies (the line 6 of the first paragraph in Results, page 2 line 13 of the first paragraph and page 3 lines 15-16 in Supplementary Note). These findings indicate that autoantibodies are unlikely involved in manifestations of the *PSMB9*-G156 mutation and that the mutation does not cause an autoimmune disease. Because the *PSMB9*-G156 mutation showed not only autoinflammation but also immunodeficiency, we proposed the term, proteasome-associated autoinflammation and immunodeficiency disease (PRAID). However, as discussed below in the responses to reviewer 3, we now suggest proteasome-associated autoinflammatory syndrome with immunodeficiency (PRAAS-ID) as a new term to indicate a novel category of autoinflammatory diseases, similar to, but distinct from PRAAS, as represented by the patients in this study.

The experimental work on the effect of the G156D mutation on the immunoproteasome assembly and function as well as the effect on the immune system is solid. The clinical documentation I can't judge so well but it seems to be adequate as well. The article is well written. I have only a few requests for control experiments and minor text corrections:

Fig. 1g: a comparison to a skin biopsy from a healthy control should be provided

A skin section of a healthy control stained with anti-p-STAT1 Ab has been added to Fig. 1g. A description in Methods (page 12) was also changed.

Fig. 3: please indicate in the legend from what kind of cells the extract has been prepared. A negative control for the ubiquitin staining in Fig. 3C should be displayed.

As shown at the beginning of Fig. 3 legend, immortalized B cell extracts were prepared. For clarity, 'Cell extracts were ..' were changed to 'The immortalized B cell extracts were ..'. A skin section of a healthy control stained with anti-ubiquitin Ab has been added to Fig. 3c.

Extended Data Fig. 1: a negative control for anti-phospho-STAT1 staining from a healthy control should be displayed in panel 1f and for the H&E staining of muscle in panel 1b a healthy control would be needed as well.

A skin section of a healthy control stained with anti-p-STAT1 Ab has been added to Supplementary Fig. 1f. Regarding Supplementary Fig. 1b, we have no available muscle samples of a healthy control due to ethical issues.

Extended Data Fig. 4: in panel 4a a healthy control is missing for the anti-ubiquitin staining. In panel b the chymotrypsin-like activity of the proteasome is measured in total lysate using the fluorogenic substrate Suc-LLVY-AMC. It has been shown that this substrate can be cleaved by cellular proteases other than the proteasome. Therefore, the peptidase activity at least for the controls should be measured in the absence and presence of a highly selective proteasome inhibitor like lactacystin or epoxomicin.

A skin section of a healthy control stained with anti-ubiquitin Ab has been added to Supplementary Fig. 4a. The proteasome activities were measured in the absence and presence of a proteasome inhibitor, Carfilzomib and shown as Supplementary Fig. 4b. The activity was decreased in IFN- γ -stimulated patient2-derived skin fibroblasts in the presence of SDS, indicating the defect of 20S proteasome. Methods in the main text (page 13) and Supplementary Fig. 4 legend have been changed.

Extended Data Fig. 5c: It has been shown that Suc-LLVY-AMC can be cleaved by cellular proteases other than the proteasome. Therefore, the peptidase activity at least for the controls should be measured in the absence and presence of a highly selective proteasome inhibitor like lactacystin or epoxomicin.

The proteasome activities were measured in the absence and presence of a proteasome inhibitor, Carfilzomib and shown as Supplementary Fig. 5c. The activity was decreased in IFN- γ -stimulated embryonic fibroblasts from PSMB9 G156D mutant mice in the presence of SDS, indicating the defect of 20S proteasome. Methods in the main text (page 13) and Supplementary Fig. 5 legend have been changed.

Introduction page 4: the definite article 'the' is needed in front of the words: Proteasome (line 7), 20S (line 9), immunoproteasome (line 13), thymoproteasome (line 14).

We added 'the' at the pointed parts.

Reviewer #3 (Remarks to the Author):

General comments:

The manuscript by Kanazawa N et al reports two unrelated patients with the same de novo missense variant in PSMB9. Both patients presented with clinical manifestations observed in previously reported CANDLE/PRAAS patients in addition to immunodeficiency. Proteasome protease activity dysfunction was detected in patients' cells. The authors generated a mouse model harboring the patients' variant in which the proteasome defect and immunodeficiency were recapitulated, although the mice did not have a clinical phenotype.

The manuscript clearly depicted proteasome function assays both in human and mouse cells.

I have the following comments below:

-Throughout the text, ideally "mutation" should be replaced with "variant" to be in accordance with recent genetic nomenclature.

Throughout the manuscript, 'mutation' was replaced with 'variant'. 'NM: not mutated' in Fig. 2a and 2b was removed.

Abstract:

"We propose the term, proteasome-associated autoinflammation and immunodeficiency disease (PRAID), as an umbrella name for our cases, PRAAS with immunodeficiency, as well as PRAAS described so far."

The sentence is not clear to me. Are the authors suggesting that PRAAS and PRAAS with immunodeficiency (ID) be a subtype of PRAID or that only the described cases of PRAAS with ID be called PRAID? In addition, as the authors also mention in the discussion, PRAID has been proposed by Poli et al as a term for "POMP-related autoinflammation and immune dysregulation disease".

As described in the response to the comment of reviewer 2, our cases are different from POMP variant cases and featured as autoinflammation accompanied with immunodeficiency. We propose a new designation "proteasome-associated autoinflammatory syndrome with immunodeficiency (PRAAS-ID)" to indicate a novel category of autoinflammatory diseases, similar to, but distinct from PRAAS, as represented by the patients in this study. We agree that it seems better to avoid the term PRAID, which is already used for POMP variant cases.

Results:

The authors start describing patient 2 instead of patient 1. I understand patient 2 had evidence of an IFN signature whereas patient 1 did not have but it would be better to start the clinical part of the results section with patient 1 for clarity.

We appreciate this suggestion. We have changed the first paragraph according to the suggestion.

I also have the following specific comments:

Page 4 Line 25: "These patients showed the common features, such as ..." I think this statement is incomplete. Did you mean "These patients showed the common features of PRAAS, such as ..." ?

We mean 'similar with each other' by this 'common'. We have changed this part to 'These patients showed similar features, such as ..'.

Page 4 Line 31: Please replace "central spinal fluid" with "cerebrospinal fluid".

This part has been changed.

Page 4 Line 34: "Patient 2 also showed manifestations compatible with hemophagocytic lymphohistiocytosis"

This part has been changed.

Page 5 Lines 4-5: "Interestingly, polyoma (BK/JC) viruses were detected in both patient's peripheral bloods."

- What treatment were the patients receiving or at what stage of their disease they were when BK and JC viruses were detected? It would be good to have a brief description of both patients' therapies in the main manuscript in addition to the supplementary appendix.

They were described in the last part of the first paragraph of Results, as follows.

' In patient 1, the viruses were detected at 4 and 10-year old, when the effects of immunosuppressants cannot be excluded, while in patient 2 they were detected at 31 days of age, at the time without any therapy.'

They were described also in Supplementary Note, the fifth line from the bottom of the first paragraph and the eighth line from the bottom of the second paragraph.

-Replace "bloods" with "blood" or with "blood samples"

We have also changed 'bloods' to 'blood samples'.

Page 5 Lines 9-12: The authors mention that the presence of pulmonary hypertension

present in their patients is a difference between them and PRAAS patients. However, in Sanchez et al, JCI, 2018, one of the 10 CANDLE/PRAAS patients reported had pulmonary hypertension. Regarding the presence of immunodeficiency, the two PRAAS2 (or PRAID) patients reported by Poli et al, AJHG, 2019, also presented with immunodeficiency.

We appreciate this pointing. As pointed by this reviewer, pulmonary hypertension is reported, but rare in PRAAS. Therefore, we have removed ‘unlike PRAAS’ and changed the parts as follows.

‘However, they apparently lacked lipodystrophy, one of the main characteristics of PRAAS, and showed pulmonary hypertension, which is rare in PRAAS.’ It seems that POMP variant cases reported by Poli et al show autoimmunity rather than immunodeficiency. This was described also in response to the first comment of reviewer 2.

Page 5 Lines 14-30: The presence of the same de novo variants in PSMB9 in the 2 unrelated patients and the functional assays indicate that PSMB9 is the best candidate gene in the patients. However, there are many similarities between the patients herein reported and patients with NEMO-deleted exon 5 autoinflammatory syndrome (NEMO-NDAS) described by de Jesus et al, JCI, 2020. In NEMO-NDAS patients, the IKBKG variants are not detected by standard WES analysis and bam files should be manually inspected for the region where these variants were reported. I think it would be important to rule out NEMO-NDAS for the 2 patients reported.

We appreciate this pointing and have checked this possibility. We re-evaluated the NEMO genes and found no variants, excluding that our cases belong to NEMO-DAS. These were described in pages 5 (Results) and 10 (Methods).

Page 5 Lines 20-22: “The latter, a rare polymorphism (rs183923514), was also detected in his father without any autoinflammatory manifestations (Fig. 2a) and IIG is not conserved even in mouse.”

Variant PSMD9 p.G11S has a minor allele frequency (MAF) of 0.0004483 (gnomAD) being more frequent in the East Asian population (0.005745). The variant is rare but is not a polymorphism as the MAF is below 1%. Moreover, since PRAAS can have a digenic inheritance mode (Brehm et al, JCI, 2015), the fact that the unaffected father carries the variant does not rule out it as candidate. I do agree that the PSMD9 most likely is not causing disease in the patient (based on conservation and MAF in East Asians) but the sentence should be rephrased.

We appreciate this pointing. The response was described in the response to the comment of the reviewer 1.

Page 5 Line 36: “... patient 1-derived cells ...” What cells? Please specify the type of cell here.

In this part, patient 1-derived immortalized B cells were used. We have made the parts clearer.

Discussion:

Page 9 Lines 6-13: "Here we would propose a new designation "proteasome-associated autoinflammation and immunodeficiency disease (PRAID)" as an umbrella name for our cases as well as PRAAS and POMP-related RAID"

Since Poli et al suggested PRAID as a term for "POMP-related autoinflammation and immune dysregulation disease", I'm not sure PRAID could be used for "proteasome-associated autoinflammation and immunodeficiency disease". The authors may want to consider PRAASID for "PRAAS with immune dysregulation".

We appreciate this suggestion and agree to avoid the term, PRAID. As described in the response to the comment on the abstract and according to the suggestion by this reviewer, we would like to suggest PRAAS-ID to avoid the confusion with POMP-related autoinflammation and immune dysregulation disease (PRAID).

Table 1: For pulmonary hypertension, PRAAS should be "rare" but not absent as Sanchez et al, JCI, 2018, reported the finding in 1/10 PRAAS patients.

In Table 1, we have changed '-' to 'rare'.

Extended Data Fig. 1: I understand the figure shows the clinical and laboratory findings of patient 1 with a normal IFN score and negative pSTAT1 staining in patient's skin biopsy, differently from patient 2 (Fig. 1). It's not clear in the manuscript text that patient 1 did not have evidence of an IFN signature. It would be important to state in which context the IFN score was measured and the skin biopsy was performed. Was IP-10 measured for patient 1?

We appreciate this comment on patient 1. Those were described in the first paragraph of Results as follows.

'In patient 1, during immunosuppressive treatments, interferon (IFN) score was not elevated and signal transducers and activators of transcription 1 (STAT1) phosphorylation was scarce (Supplementary Fig. 1e,f).'

They were described also in Supplementary Note as follows.

'... at 10-year old during the treatment with dexamethasone, expression of ISGs was not upregulated in the whole blood (Supplementary Fig. 1e) and p-STAT1 staining was also faint in the lesional skin, which was obtained at 7-month old during the treatment with prednisolone (Supplementary Fig. 1f).'

We have also added the information in Supplementary Fig.1 legend.

IP-10 level was elevated in patient 1 (line 7 of the first paragraph of Results (page 4) and 418 pg/ml, page 2 line 22 of the first paragraph in Supplementary Note), although lower than that of patient 2 (1000 pg/ml, page 3 line 25 in Supplementary Note).

REVIEWERS' COMMENTS

Reviewer #1 (Remarks to the Author):

Thank you for your clarification

Reviewer #2 (Remarks to the Author):

My suggestions have been adequately addressed both experimentally and by text modifications.

Reviewer #3 (Remarks to the Author):

The manuscript by Kanazawa N et al reports two unrelated patients with the same de novo missense variant in PSMB9. The authors demonstrated that the de novo variant leads to proteasome dysfunction and is responsible for the patients' clinical phenotype, characterized by autoinflammation and immunodeficiency. Upon review of the revised manuscript files, I conclude the authors addressed all my comments and suggestions properly.